# Smallholder Views on Chinese Agricultural Investments in Mozambique and Tanzania in the Context of VGGTs

**Rebecca Pointer** [1,*], **Emmanuel Sulle** [2,*] **and Clemente Ntauazi** [3]

1   Independent Researcher, Cape Town 7945, South Africa
2   The Aga Khan University, Arusha Campus, Ngaramtoni ya Chini, Arusha P.O. Box 499, Tanzania
3   Institute for Poverty, Land and Agrarian Studies, University of the Western Cape,
    Cape Town 7535, South Africa
*   Correspondence: pointerrme@gmail.com (R.P.); emmanuel.sulle@aku.edu (E.S.)

**Abstract:** Based on a case study in each country, this study documents the views of Mozambican and Tanzanian smallholders regarding Chinese agricultural investments and the extent to which investors abide by their legitimate land tenure rights as defined by the *Voluntary Guidelines for the Responsible Governance of Tenure of Land, Forests and Fisheries in the Context of National Food Security* (VGGTs). The VGGTs offer guidelines to government on how to protect the land tenure of rural communities when land is being acquired for large-scale land investments. The study also assessed the impact of the COVID-19 pandemic on smallholders. Due to COVID-19, instead of fieldwork, we conducted telephone interviews with 20 smallholders in Mozambique and 35 in Tanzania. The Mozambican case showed that even when land set aside for investors was not in dispute, smallholders still had unmet expectations, especially regarding investors' corporate social responsibility activities. In the Tanzanian case, even though the land leased by the Chinese investor had been designated as general land, it had laid fallow for a long period, and smallholders had moved back onto the land, only to be displaced in 2017. Although smallholders' views on the investment were mixed, the case underscored the need for government to assess current land use before allocating it to investors—regardless of how the land is classified and especially in areas where land shortages are creating conflict. The cases show that even if communities are consulted about proposed land investments, guidelines need to include clauses that allow for ongoing communications between investors, communities and government officials such that if communities are unsatisfied with the results of the investment, renegotiation is possible. Further, in the event of crises, such as COVID-19, investors should partner with communities and government to limit the extent of harm in communities as a result of the crisis.

**Keywords:** agricultural investment; VGGT; legitimate tenure; corporate social responsibility; land tenure; Chinese investments

## 1. Introduction

Foreign land-based investments in Mozambique and Tanzania, as in many other developing countries, peaked in the mid-2000s and onward following the "Triple F" crisis (food, fuel and finances) [1]. Like other investments in agricultural land, Chinese investments sometimes create tensions between investors and communities due to the way investments affect the tenure rights of local landholders [2], even though the host countries may need investment to boost food security and create jobs and incomes.

This study documents Chinese investments in agricultural land from the perspective of smallholders in Mozambique and Tanzania, especially the extent to which smallholders believe their legitimate land tenure rights were protected by the *Voluntary Guidelines for the Responsible Governance of Tenure of Land, Forests and Fisheries in the Context of National Food Security* (VGGTs), endorsed by the Committee on World Food Security in May 2012. The VGGTs advocate for countries to undertake well-articulated land reforms to ensure

they can achieve food security and nutrition for all and support sustainable efforts to eradicate hunger and poverty. The VGGTs are intended to secure sustainable well-being, environmental protection and the socio-economic development of all people, especially those with legitimate tenure rights.

In both Tanzania and Mozambique, Chinese investments in farmland date back to the 1990s. The good market prospects of sisal were identified in 1999, and the "Kingdom of Sisal" was the first choice for Chinese investment. With the support of the Export-Import Bank of China (China Exim Bank), Chinese investors began providing technical support for sisal farming. The relationships between China and African countries are typically supported by elites, "with civil society not being able to catch up with the pace that elite ties develop" [3]. In part, this is because elites may benefit directly from such investments through rent, but they also claim that such investments strengthen the relations between host countries and China [4,5]. Therefore, even in areas where communities have demanded the return of idle land that Chinese investors have rented, land has not been returned [6–10].

In African countries, limited data is available on who owns or has tenure on which land due to several factors, especially poor information management systems, shortage of land surveyors employed by government, politicians having a poor grasp of what is needed for land management, and governments not having spatial plans for determining how to allocate land [11,12]. Therefore, in areas designated for large-scale land-based investment, it is often unclear whether legitimate tenure rights exist and for whom. Often, land is designated as state/public land, but in keeping with the VGGTs, this does not exclude the existence of pre-existing, legitimate (e.g., customary) tenure rights for community members [13]. Legitimate tenure rights, especially those of women, are often not respected when governments ignore VGGT principles and only abide by national laws and consider legal rights [14]. Further, some land-based investments are for non-food crops, which may be detrimental to the food and nutrition security of local communities/people, who may find they have less land for growing food in their local area [7,15].

This study examines whether in-country tenure regimes and VGGT principles are being considered or ignored in the case of two Chinese agricultural investments, one in Mozambique and one in Tanzania. Both Tanzania and Mozambique have dualist land laws, meaning that some land tenure regimes are recognised in their constitutions and land legislation while other categories of communal/community land are governed by customary law [1,16,17]. Based on these dualist systems, we begin by outlining the legitimate tenure rights based in each country's legislative and policy frameworks and the strengths and weaknesses thereof. We then explore how land-based investments are promoted and governed, explaining each country's policies and procedures by which investors may acquire land, especially community/communal land. Thereafter, we look at the two case studies to establish differing views on the impact of Chinese agricultural investments on the tenure of smallholder farmers in Tanzania and Mozambique. We also explore what impact COVID-19 had on small scale farmers in the case study areas, especially in terms of impact on tenure. Typically, agricultural issues experienced by rural communities under COVID-19 were access to inputs, access to markets, and access to food [18–23]. Each of these are addressed in the literature review and, where relevant, in the case studies.

Due to COVID-19, extensive field work was not possible, so stakeholder interviews were conducted by telephone with smallholders who lived near Chinese investment sites (20 interviews in Mozambique; 35 in Tanzania). While governments have earmarked vast areas of land for investment in these two countries, only a few Chinese investments exist. Therefore, in this study we chose one Chinese entity per country based on the area of land secured for investment and the business model used by investors. Data was analysed by identifying themes emerging from the literature review and interviews.

## 2. Literature Review

Both Mozambique and Tanzania have progressive land laws that provide for and protect the security of tenure for customary land holders in Africa [16,24]. However, land

tenure in Mozambique and Tanzania is increasingly influenced by the countries' respective moves towards market liberalisation, and the language of modernisation is used to promote large-scale land-based investment. Below, we examine the changing land laws and policies as well as the investment policies and procedures in the case of each country.

*2.1. Land Laws and Investment Models in Tanzania*

2.1.1. Current Land Laws in Tanzania

The National Land Policy introduced in 1995, the Land Act No. 4 of 1999 and the Village Land Act No. 5 of 1999 are the key laws governing land tenure in Tanzania [25]. Under these laws, all land types (general, reserved and village) were placed under the trusteeship of the President, and smallholders were given "deemed rights of occupancy" in an effort to remove the dualistic land rights systems that have persisted since colonial times. Reserved land, which is set aside for conservation, makes up 30–40% of Tanzanian land, including national parks, game reserves, and marine reserves. The Wildlife Conservation Act of 2009 and the Forest Act of 2002 guide the use of reserved land [26].

Village land is that which has been demarcated or agreed to be within the boundaries of Tanzania's more than 10,000 villages, as defined by local government legislation in the 1970s and 1980s. In the Village Land Act, village land falls under the authority of 25-member Village Councils, drawn from representatives of political parties chosen by community members over the age of 18; at least one third of the Village Council must be women [25]. The Village Council must consult with Village Assemblies before decisions are made about land allocation and land investment.

The general land means all public land that is neither reserved nor village land, except unused village land. Commissioner of Lands in the Ministry of Lands, Housing and Human Settlements Development holds authority over general land [25]. With respect to general land, the Land Act specifically intended to create a land administration framework to "facilitate making land available for private or foreign investment". More than 2.5 million hectares of general land have been identified as suitable for investors and this land is organised into a "land bank" under the authority of the Tanzanian Investment Centre (TIC). Much of the land earmarked by the "land bank" had weak recognition under customary and communal land rights [26] and mostly consists of abandoned former estates, which had previously been alienated from villages. Even though the land was still designated as general/estate land, once the land was no longer in use, villagers returned to the land. This has led to disputes between investors and communities, and between communities and government, as to whether the villagers occupying the land have the right to secure tenure.

Furthermore, "unused land" was often being used for various non-agricultural purposes, such as livestock grazing or accessing tree and non-tree forest products [25]. Despite the ways communities were using the land, investment programmes went ahead with the erroneous assumption that investors could easily access unused prime land. Under these tenure regimes, consecutive Tanzanian governments prioritised investment in tourism, agriculture and livestock and gave investors tax breaks. Soon, smallholders and pastoralists were experiencing land acquisitions from individuals who encroached on their land using "questionable methods" [27], with "poor standards of accountability and transparency [ . . . ] [allowing] officials to cash in on, for example, user fees, exclusive licenses and land allocations".

2.1.2. Investment Models in Tanzania

Land-based investments in Tanzania are mainly guided by the Land Acquisition Act of 1967 and the Tanzania Investment Act of 1997, which established the TIC and laid out procedures for foreign investors to acquire land in Tanzania [26]. These procedures were then further strengthened in the Land Act of 1999 and Village Land Act of 1999, which provide the legal and procedural framework to guide how to acquire, transact and dispose of land. However, the framework has some ambiguities in terms of how these rights are protected during investment processes [28,29]. The existing three types of land (general

land, village land and reserved land), despite sharing some commonalities, are governed by different rules, sometimes creating conflict.

For example, the Land Acquisition Act of 1967 has not been reformed and allows the President to transfer village land into general land with the approval of the village authorities as provided for in the Land Acts [26,30]. The Act outlines how land can be compulsorily acquired for "public purposes", including for investment, and the amount of compensation to be paid to affected individuals or communities. However, the Land Act and Village Land Act of 1999 do provide some safeguards, as they lay out democratic processes and structures for governing land at national and grassroots level.

For example, any investor or company that wants to acquire village land for investment purposes must meet one of two key eligibility criteria: (i) they must have a minimum investment capital of USD 100,000 if fully or majority-owned by Tanzanian citizen(s), or (ii) USD 300,000 if fully or majority-owned by non-Tanzanian citizen(s) (this also includes companies incorporated under the laws of any country other than Tanzania) [26]. Foreign investors or companies cannot directly purchase land and must instead work with the village government and Commissioner of Land to have such land transferred to the category of general land to extinguish pre-existing rights. Depending on the nature of investment and other requirements for investment returns, the investor then needs the Commissioner of Land's approval to receive a special title deed known as a Granted Right of Occupancy (GRO) for 33, 66 or 99 years or derivative rights, which are provided for 32, 65 and 98 years [26,31].

Domestic investors may proceed without approaching the TIC for an investment certificate, but foreign investors who meet the basic capital requirements must introduce their business idea to the TIC, with relevant documentation including, inter alia, the business plan and feasibility studies and audited accounts for the past three years [26]. Once they fulfil TIC requirements, they may be registered, verified and given a "Certificate of Incentives". The certificate grants the investor a license to use the TIC one-stop point for social services, such as immigration services, labour services, the Tanzania Revenue Authority (TRA), services of the Ministry of Lands and Human Settlements, Tanzania Bureau of Standards (TBS), Business Registration and Licensing Authority (BRELA), National Environment Management Council (NEMC), Occupational, Safety and Health Authority (OSHA), Tanzania Food and Drugs Authority (TFDA), and Tanzania Electric Supply Company Limited (TANESCO). The certificate further lists all investment favours, including tax holidays. The certificate and other introduction letters are then sent to lower levels of government, such as districts and villages, and the investor then introduces their interest in land at the village level.

If they are welcomed in a district, the foreign investors must follow procedures to secure the transfer of village land to general land [26]. For example, the investor must ensure those occupying the land or having been allocated the land previously are given written notice by the village council. For transfers of less than 250 hectares of land, the village council must submit recommendations to the general village assembly, who may then refuse or accept the proposal based on information and advice provided by the district council, as provided by the Village Land Act No. 5 of 1999. Currently, village land transfers are the dominant way of obtaining land, both for small/medium-scale and large-scale land-based investments.

For transfers of more than 250 hectares, the Minister of Lands must consider the recommendations of the village council, general assembly and district council before granting permission for village land to be transferred to general land (Village Land Act, Section 4 (6)) [26]. Once a local investor has received a GRO from the Commissioner for Lands or a foreign investor has received the Certificate of Incentives from the TIC, the Commissioner may issue a Letter of Offer to the applicant, which outlines the terms and conditions of their land access. The investor then accepts the Letter of Offer by paying a required fee and filling in a prescribed form, agreeing to a premium or annual rent.

Once the procedures are satisfactorily concluded, the Commissioner of Lands will issue a Certificate of Occupancy in the name of the President.

For both local and foreign companies, general land is somewhat easier to access, as an investor obtaining land from the TIC "land bank" does not need to negotiate with villagers [26]. As such, the investor can complete the process to secure rights within 30 days of submitting their application. Investors willing to secure general land on their own initiative can also access land from a company or an individual person wishing to sub-grant an existing right or sell their granted rights. Under the law, any holder of a granted right may sell it to any willing buyer, but any action to transfer a right over land is subject to the Commissioner of Land's approval [26,31]. In Tanzania, since the colonial era, several companies and individuals have had GROs for estates and plantations on general land. When a company holds the granted or derivative right, the law provides it the right to occupy and use land and the right to exclude others' access to and use of the land. Despite some companies holding granted or derivative rights to estates, some have either ceased production or abandoned the land after nationalisation; nearby villagers then opportunistically occupied this land as it was not being used.

*2.2. Land Laws and Investment Models in Mozambique*

2.2.1. Current Land Laws in Mozambique

Since independence in 1975, land in Mozambique has been the property of the state, and the selling, mortgaging or encumbering of land is not constitutionally permitted [24]. The overall responsibility for land administration and cadastre lies with the National Directorate of Land (DNT) in the Ministry of Land, Environment and Rural Development, which allocates provincial and district services (decentralised). At a national level, the DNT is the regulatory authority, charged with holding and organising the national land cadastre records and, in the case of large-scale land applications over 1000 ha, responsible for processing applications. The DNT also provides technical guidance to the cadastral services of the provincial administrations and the decentralised municipalities. For rural land, the Provincial Service of Geography and Cadastre (SPGC) has primary operational responsibility [32].

The 1997 Land Law defines limited land use rights for occupants and users on the basis of a unitary system of tenure known as the *Direito de Uso e Aproveitamento da Terra* (DUAT), i.e., the right to use and benefit from land [33]. The DUAT is at the heart of community land rights in Mozambique and can be obtained via three channels: (i) customary tenure rights; (ii) individuals who have occupied land in "good faith" for at least 10 years and who can prove land rights through testimony without registering or titling the land; and (iii) individuals can apply for a DUAT for a particular piece of land for up to 50 years, with one renewal, and a land rights concession, typically for natural resource extraction or developing agricultural, forestry or fishing activities [33,34].

While community members can obtain a DUAT by occupying land for 10 years, individuals requiring land for non-housing or non-community purposes must apply for a DUAT title. The government grant does not set minimum or maximum sizes of land that can be allocated. However, the grant applicant must prepare a DUAT use plan, which the state then evaluates and can issue a provisional grant for two years [35]. While this type of DUAT is seen as an effective way to secure land [36,37], critics warn that these titles are sought for land speculation and prevent the poor from securing land [38,39].

The long-standing occupation and management of land confers communities with land rights, without the necessity to register that land [40]. Under Article 10 of 1997 Land Law, communities can be regarded as title holders based on the DUAT if they can offer proof of land rights through oral testimony, eliminating the obstacles of surveying, registering, and titling that often prevent the poor from securing their land rights [24]. Hence, rural communities are not required to hold a formal DUAT title, but their DUAT rights are legally recognised and protected [32,33].

A DUAT can be conferred to singular or collective subjects, taking into account its social and economic purposes [41]. For economic activities, a DUAT can only be conferred for 50 years, so investors are only entitled to use the land for 50 years [24]. Investors are allowed to renew their DUAT at the end of this 50-year period by submitting a new application. The basic provision of land law states that the land is the property of the state and cannot be sold or otherwise alienated, mortgaged or encumbered. It also reasserts that individuals, communities and entities have the right to use and benefit from land for a long-term or perpetual period.

Since the introduction of the 1997 Land Law, the Mozambican government has been promoting smallholder family farming under the National Development Programme (PROAGRI); however, the state only gave 4% of its annual budget to developing agriculture from 2000–2008, 8% in 2010–2011, and 6.5% in 2015–2017, so smallholder agriculture remains marginalised [5,42]. Currently, more than 90% of land in Mozambique is under unregistered good faith occupancy and customary tenure [43]. This is a key source of land tenure risk because it creates problems regarding the visibility of rights, which often leads to the erroneous (or disingenuous) interpretation that land is available when this is actually not the case. Without a formal DUAT, local governments and investors often fail to adequately recognise community land rights and uses, leaving both communities and investors at risk [44].

Communities may seek a formal DUAT and can then go through a formal community consultation and mapping process, called "delimitation" [45]. Thereafter, they will receive a certificate and may mark their land with place markers around the perimeter; they may then "apply for a formal land title" [40]. This procedure is recommended, as it confers communities with a legal personality in concrete form, so they can enter into third-party contracts and open a bank account [24,35]. This process has been adopted in some instances where an investor is interested in accessing community land. However, few communities or individuals have the resources needed to apply for and receive a formal title, since the process is cumbersome, time-consuming and prohibitively costly for many. Further, the institutional capacity of local land authorities to survey land and register DUATs is limited [46].

Revisions to the 1997 Land Law have also affected community rights [24]. For example, Article 27 gives decision-making power to the Local Consultative Councils created under the 2003 Local Government Bodies Law. Further, Article 35, increases state control over community rights, and communities must now submit a land use plan before they get the go-ahead to demarcate the land [24,47]. Because of this decree, the process by which communities can secure a formal community DUAT has ground to a halt, and communities are left in an administrative limbo, having delimited their land but having received no certificate. The 2011 Ministerial Diploma (No. 158/2011 of 15 June) also impacted land tenure, as it specified how consultations should be conducted but provided less scope for community consultation. In addition, a new Constitution was adopted in 2004, which altered some community land rights [47]. For example, while the 1990 Constitution grants DUATs based on "social purpose", the new Constitution grants DUATs based on "social *and* economic purpose". While the 1990 Constitution powerfully stated that land allocation would prioritise direct users and producers, and that the law would not permit situations in which economic privilege could be used to the detriment of most citizens, this clause was removed from the 2004 Constitution.

### 2.2.2. Investment Models in Mozambique

Land-based investments in Mozambique mostly involve national, foreign companies or individuals acquiring land rights through long-term leases or concessions. Another form of land-based investment involves contract farming programmes with smallholder farmers [48]. These investments are promoted by the investment promotion centre, the *Agencia de Promoção de Exportações e Importações* (APEIX), which plays a key role in ensuring that the country is competing with other countries to attract investors to different sectors.

Despite the government's efforts to roll back some of the substance of the 1997 Land Law, changes have been met with resistance, such that by 2009 a moratorium was introduced on land deals involving more than 1000 ha of land [49]. The moratorium gave the government time to scrutinise large-scale agricultural investments and alter their approach. As a result, few large-scale investments have taken place since 2010, suggesting "a more cautious and nuanced approach to agricultural investments". For example, in 2008/9, 85% of DUAT applications were approved, while in 2010–2012 only 29% were approved.

Land-based investments in Mozambique are affected by several policies, legislation and strategies. However, at the time this research was undertaken, the main legal frameworks were provided in the 1997 Land Law, the Strategic Plan of Development of Agriculture Sector (PEDSA) (2011–2020), and the National Agriculture Investment Plan (PNISA) (2014–2019). Apart from the domestic legal framework, international guidelines such as the VGGT [13] and the Committee on World Food Security Principles for Responsible Investment in Agriculture and Food Systems (CFS RAI) [50] are also taken into consideration, although the Ministry of Agriculture and Rural Development does not consider it of great importance to land-based investment in the country.

Investors can access land through community consultation, which involves the applicant (investor), the community, the District Administrator, and representatives of the Provincial Services of Geography and Cadastre [47]. During the consultation, the investors must provide details of the project plan, and communities should have the right to ask questions about the investment and its impact on their livelihoods. Meeting minutes must be taken, and all parties must sign the minutes, which serve as a record of agreement. For areas smaller than 1000 ha, the provincial governor must sign off on the deal; for areas of 1000 ha–10,000 ha, the Minister of Agriculture must approve the deal; and for an area greater than 10,000 ha, only the Council of Ministers may approve the deal.

The key policy instrument for agricultural investment is the National Agricultural Investment Plan (PNISA), which aims to increase food production and productivity to fight hunger, reduce poverty and eliminate food insecurity [51]. PNISA defines key guidelines and strategic actions for a coordinated government intervention that may allow the rebirth of agro-industry in Mozambique. It seeks to guide public and private investment, including from international development agencies, so as to promote agro-industry.

Government policies and strategies for agricultural development seek to increase production and productivity by (i) using agricultural inputs and technological packages, (ii) providing access to markets for producers and vulnerable groups, (iii) improving food security and nutrition, and (iv) promoting sustainable and rational use of natural resources [51]. However, PNISA does not statistically consider smallholder farmers, who largely produce staple food crops, and gives greater priority to the private sector and middle- and large-scale commodity producers, which is touted as the master formula to address food insecurity and hunger.

Combining the rights of investors under long-term concessions and communal tenure is a difficult task, given that different authorities use different codes [52]. In recent years, many land-based investments have taken place without public consultation, thereby giving investors space to operate without taking into account community-friendly, environmental and social values [53]. Despite the national legal framework protecting peasant rights, domestic elites still have access to and control of the land, which can ultimately lead to communities being dispossessed [54]. Power imbalances exist between various levels of government and the rights-holding communities, and investors typically leverage one of these power-brokers to acquire land, including Customary Authorities, Community Authorities and District Administrators, politically connected elites, and professional networks. Below, we highlight how these groups are implicated in giving investors access to land without proper community consultation.

*Customary authorities* have decisive power in consultations between communities and investors [54], and sometimes, especially in the central provinces of the country, authorise projects without communities' free, prior and informed consent [54]. Since investors

often assume that consulting with customary authorities is equivalent to consulting the community, communities may not benefit from the deals because they have not been sufficiently consulted.

Similarly, *Community Authorities and District Administrators*, including traditional leaders, village secretaries and leaders legitimised by communities [24], make final decisions about deals without sufficiently consulting communities and sometimes adopt rent-seeking behaviour in the form of "facilitation fees" [47]. When conflicts arise, community authorities and district and provincial administrators are responsible for resolving them, even though they are often responsible for signing away community rights [40].

*Political elites* (war veterans, ministers and other government officials) also influence the actions of land administrators by calling in political favours [47]. Many of these elites were awarded DUATs at the end of the civil war in 1992, and even if their land is not productively used, it is difficult to revoke their land rights due to their political power.

*Professional networks* of Mozambicans with business connections and business know-how have also become involved in land deals [47]. They use their knowledge of local markets to leverage financial gains, either through partnerships with international investors or by safe-guarding deals on behalf of investors to ensure they are not exposed to significant financial risk.

Despite these various role-players moving to access land without community consultation, communities can leverage an Environmental and Social Impact Assessment (ESIA)—a legal requirement for land investments—to limit or prevent land investment [55]. The ESIA process is administered by the Ministry of Land and Environment and conducted by an environmental consulting firm chosen by the investor. The ESIA must identify, predict and evaluate both positive and negative potential impacts and identify mitigating and compensatory measures to minimise negative impacts and optimise positive ones [56,57]. A formal and informal, oral and written public participation process is then applied to review the ESIA in relation to (i) the accuracy of the data and the elaboration of the environmental diagnosis; (ii) identifying impacts not previously brought to light; and (iii) identifying which impacts are of greatest concern to the public [55]. Thereafter, the ESIA and the results of the participatory process are submitted to a review committee in the National Directorate of Environment Assessment (DNAIA). Reviewers may request additional data to supplement specific studies to support decision making, and if the project is deemed viable, an environmental license will be issued, allowing the investor to proceed with the project.

Despite these measures, some ESIAs have been presented as valid and scientific but have failed to address cumulative impacts [58], especially those that affect the livelihoods of local communities. Further, they have not highlighted any weaknesses in their analysis or the level of confidence they have in the findings [59,60]. The relevant documents also tend to be inaccessible to the public, since the government argues that they are confidential; it is therefore difficult for CSOs and outside experts to question the feasibility of and real motivations behind such projects. As such, while ESIAs might result in a cancelled project, this is not often the case.

The land administration system is also weak due to relevant authorities often being over-burdened and under-resourced [24,61,62] or sometimes ignoring the system altogether [40,45]. Different individuals or authorities may influence land provision to investors for their own benefit, and changes to the Land Law of 1997 and the Constitution have led to rollback of some rights [24]. Unlike Tanzania, Mozambique does not have a definitive system for investors to acquire land; as such, individuals with the relevant contacts can influence whether or not land is provided.

### 2.3. Impact of COVID-19 on African Agriculture

During the COVID-19 pandemic and the resultant government restrictions on movement to slow the spread of disease, access to food was a significant challenge for the world's population and especially in Africa [18]. Alongside food access, the livelihoods

of small-scale food producers were also negatively impacted, even though many African governments created social protection schemes for smallholder farmers and small-scale agribusinesses. Two of the key factors that created negative impacts for smallholders were the lack of transport (including transport of agricultural inputs) and the shutdown of many markets such as restaurants, marketplaces, supermarkets and exports, resulting in smallholders not being able to earn an income from sales [18–21]. Only a few farmers' collectives/associations developed responses to these market disruptions, partly because collective meetings were banned [63]. In Mozambique, food insecurity soared [18], and it became difficult to access farm inputs due to limitations on imports [21]. Even in Tanzania, where measures against movement were relatively limited compared to other countries, food quality and quantity triggered lower food consumption in some areas, though other areas experienced no disruption to food supply [19,22,23,64]. Many farmers who commonly exported products were faced with the inability to transport goods across borders, with about 85% of farmers experiencing income reduction, albeit smallholder subsistence farmers were somewhat less affected [19,64–66]. Nevertheless, the extent to which smallholders across Africa were affected by the pandemic relates to a range of contextual factors and some individual- and farm-level factors [67].

## 3. Methods

Based on an extensive literature review, including grey literature and websites, we identified two case studies for in-depth investigation—the China Africa Development Company (CAD) in Mozambique [68] and the China State Farm Agricultural Company in Tanzania [69–75]. In the literature review, only five other Chinese agricultural investments were identified in Mozambique and six in Tanzania, though some of these were no longer operating [6,69,72,73,76–91]. The two case studies were selected because they revealed the complexity of large-scale land investment and the different approaches of Chinese businesses when farming large estates.

### 3.1. Data Collection

Initial fieldwork was undertaken in Tanzania in October 2019, but due to elections in Mozambique, field work did not begin until 2020. Again, due to the COVID-19 pandemic, it was not possible to travel to the two case study sites in 2020, so fieldwork relied on telephone interviews with community members and members of farmers' associations/organisations. We made initial contact with farmers' associations in each area, which we had identified through the literature review. We then used a snowball sampling method to talk to smallholders—whether they were part of the relevant farmers' association or not— as well as other community members and agro-dealers. We conducted 20 interviews in Mozambique and 35 in Tanzania. The focus of fieldwork was collecting the views and perceptions of interviewees regarding the impacts of the Chinese investment in that area, including any problems and benefits. We used an open-ended questionnaire, shown in the Appendix A to this article.

Because the pandemic was affecting people across the world, an addition was made to the project to gather the views of community members and members of farmers' associations/organisations about the impact of the pandemic on relationships with the case study business, and more generally, the impact of the pandemic on smallholder farming and land tenure.

### 3.2. Analysis

Analysis was then undertaken by identifying themes emerging from each case study, highlighting similarities and differences.

## 4. Results

### 4.1. Mozambican Case Study: China Africa Development Company

The China Africa Development Company (CAD) has operations in Matutuine, the southernmost district of Maputo province in Mozambique [68]. Unlike other large-scale agricultural investments operating in the country [7,92,93], CAD, with local government support, negotiated with local smallholder producers to acquire land through the *Associação de Camponeses de Salamanga*, a smallholder farmers' association in Salamanga village (hereafter referred as the Association), which currently holds the land rights. Like other associations in Mozambique, the primary objective of the Association was to secure inputs from the government and NGOs and create a space for smallholder producers to share local knowledge and help one another in the production process [94]. Association members produce horticultural products, such as cabbage, onions, tomatoes, lettuce, eggplant, parsley, and dry land crops like maize, potatoes and ground nuts; small family units using manual production methods are typical in the area [95]. However, production has been uneven, and crops have sometimes failed due to the low fertility of the land. CAD acquired land to produce similar crops but has also been growing rice to the extent that the Matutuine district is leading rice and vegetable production in Maputo province [96]. These successes contribute to food availability and enhance food security in the area.

CAD also employs 30 Chinese nationals, including five women, and 29 people from the communities around the company investment area; it also hires 70–100 seasonal workers as needed. Company policy dictates that they select labourers with low education levels to keep wages down, but it has still significantly contributed to providing local labour and has reduced unemployment in the district. The negotiation between the company and the Association started in early 2011. While some local farmers chose to focus on improving the soil fertility of their land through crop rotation, others granted part of their land to CAD so that the Chinese technology and equipment could ameliorate land salinisation. According to one Association member, interviewed on 20 September 2020:

> The Chinese approached us wanting land for investment. Our association has a huge amount of land, more than 150 ha, and we were only exploring 20 ha because part of it was salinised, and therefore, unproductive.

According to the Association member, a Memorandum of Understanding (MOU), developed by the local government was then set up, under which the Association agreed to grant CAD 150 ha for 30 years. The MOU includes clauses that specify that during the 30 years, CAD would provide farming inputs like seeds and fertilizer and would plough the farmland of each Association member without extra charge, except fuel. However, according to interviews with community members, the MOU was signed without free, prior and informed consent (as provided for in VGGT TG No. 3). They said no participatory process was undertaken to collect members' opinions and clarify some clauses; for example, the MOU contained no termination or indemnification clauses. Therefore, some local farmers argue that this was a top-down approach, and they felt that the local government failed to protect the interests of smallholder producers and smallholder farmers, who do not benefit from the deal.

Now that the MOU has been in place for more than ten years, some changes have occurred within the Association: in-depth interviews revealed that the Association, which initially had 50 members, now has less than 10 members due to conflicts. Interviews conducted with Association members, agro-dealers and individual smallholder farmers in Salamanga village revealed that although some were sceptical about the deal and feared CAD had a strategy to accumulate more land and water resources over time, most conflicts were not about land tenure rights. Where concerns were about land, they arose because the Association does not have a formal DUAT, and CAD had said it intended to acquire a DUAT in the near future. Therefore, some members feel land insecure, foresee a high probability of land acquisition by the company and therefore likely fear the loss of community land.

More pressingly, according to those interviewed, conflict among members arose because the MOU conditions were not binding. For example, although in the MOU CAD agreed to plough members' farmland for free, this did not take place. Other members criticised the local government agriculture department, District Services of Economic Activities, for not assisting them as needed, resulting in CAD farm inputs and tractor hire being too expensive and therefore unaffordable. One explained that, excluding the money needed to buy fertiliser and seed, "the cost for ploughing land is 160,000 MZN [USD 21.32] per hour. One hectare requires two hours, amounting to 320,000 MZN [USD 42.64]". However, a farmer needs to use the tractor three times to get the field ready for sowing and production, so the total costs is 1,900,000 MZN (USD 253.18).

Further, according to some interviewees, CAD has failed to transfer, share or exchange production knowledge or skills with local smallholder farmers, despite the MOU and expectations of many smallholder farmers. The seeds used by the Chinese company are imported from China, which means it is not open to, or associated with, local seed markets. All production capital and consumable inputs, including seeds, fertilizer, tractors and other equipment are imported from China, which has resulted in local agro-dealers and some associations complaining that the company has not led to them extending their market. As an Association member explained in an interview on 25 September 2020:

> We don't share seeds or even receive training from CAD. We have requested this several times, but it seems that we need government to talk with them; they don't hear us.

Farmers said that attempts to directly approach the company have not led to the expected level of support; instead, the Matutuine district government has intervened several times and tried to persuade CAD to undertake CSR projects.

CAD sells most of its produce on the national market but mainly to Chinese citizens, including those working for Chinese supermarkets, restaurants, hotels and companies. A small portion of the total output is sold at local agriculture trade centres in Bela Vista and Salamanga; some is also sold in local markets. According to interviewees, further conflict occurs because opinions differ widely with respect to CAD selling some of their produce in the local market at cheaper prices. While many interviewees preferred the lower prices of rice, many also complained about the quality and said that Chinese supermarkets received better quality rice than they did at local level. Those who preferred the lower price felt that the difference in quality was understandable as the company could secure higher prices for the better-quality produce from Chinese buyers, while produce of lower quality—and thus lower price—was more attractive locally. However, making cheaper CAD rice available at local markets also led to some local vendors losing customers, creating conflict and unhappiness.

Although many criticisms have been levelled at the company, some community members are still pro-investment. Farmers who said they had benefitted from investment in the area pointed to the collateral side-effect of the company opening the valleys to access water (as its water usage is high), which resulted in the community now having access to more land. In addition, infrastructure developments, promoted by the company to maximise and optimise its own profits, have also resulted in advantages to the community, albeit in asymmetrical ways [94,97].

With respect to COVID-19, the preventative measures imposed by the Mozambican government restricted movement to a point that "partly exacerbated existing socio-economic inequalities among food system actors" [98]. Agricultural inputs became difficult to access and more expensive due to such transportation problems. While local NGOs and local government authorities tried to address the seed price crisis by supplying some seed varieties to smallholders for free [18,99], this was insufficient to cover all smallholder seed requirements.

To address COVID-19-related production challenges, the provincial government also provided an irrigation system to 53 smallholders in the area (41 women) [99]. However, due to restrictions on movements and closures of businesses such as restaurants, those growing commercial crops could not reach markets and therefore had to sell produce locally at lower

prices [18,98]. More could therefore have been done to transport food safely so that it could reach markets in a timely manner [98]. Some smallholders entered an agreement with a Spanish NGO—Asociación CESAL—who supported the smallholders in marketing their produce, thus slowing impact [100].

While impacts on tenure security were felt by some communities across Mozambique, due to government moving ahead with registering investor land without community consultation [18,101], we did not find any evidence of this in terms of CAD.

### 4.2. Tanzanian Case Study: China State Farm Agribusiness Corporation (Tanzania) Ltd.

The China State Farm Agricultural Company in Tanzania is located in Kilosa District—one of the seven districts of the Morogo Region in South-East Tanzania [72]. With the aim of growing sisal, the China State Farm Agribusiness Corporation (Tanzania) Ltd. acquired 6900 ha in the area (for 99 years) at a cost of USD 1.2 million, but it has only been using one fifth of the area [71,72]. Half of the sisal is marketed to local (non-Chinese) biogas factories, and the remainder is marketed to buyers in China and the Middle East [72,102,103]. Yields have been steadily growing, with resultant profits [71]. The farms employ about 1000 workers, and between 200 and 300 workers live on site in free accommodation [75]; the company also has a complement of 100 senior- and mid-level management, only a few of whom are Chinese [70,72,102]; however, it is unclear how many managers are from the surrounding area.

Although conflict in Kilosa has been declining in recent years, it has been the site of many conflicts between farmers and pastoralists due to land tenure policy and laws and agricultural policies, rampant corruption among public officials, an inefficient dispute settlement framework and shortage of land to meet the demands of the competing users [26,104,105]. The decline in conflict may be attributed to the district government having established a village peace committee to settle disputes between farmers and pastoralists; further, village land use planning has been undertaken, and several NGOs are undertaking land titling in the area [106].

However, despite the company having operated in the area for many years, many community members are unclear as to how the company acquired the 99-year lease and by what means the government decided in 2017 to evict them from the company's unused land, on which they had been farming. Villagers felt that a key reason behind land shortages is that large estates had been established in or near their villages and large pieces of this land were unused. As one villager explained in an interview on 5 November 2020:

> *We used the said company's land due to scarcity of village land planned for farming, we wished the company could release land which is not suitable for sisal production to the village to be used for farming, especially the wetland, but the company and authorities do not seem to hear us.*

Nevertheless, many villagers also said employment offered by the company had increased their incomes and standard of living, especially those employed to plant, tend and harvest sisal. As explained by one interviewee on 5 December 2020:

> *This Chinese company gives priority to our villagers when they hire casual labour, they usually bring employment opportunity notices to our village office when they need casual labourers, and those* [village members] *who are interested in working for the Chinese company are given the first priority.*

However, employees also complain of tough rules, with long working days and their productivity being closely monitored; if a Chinese supervisor says that they are not meeting their quotas, they risk having their salary reduced.

Nevertheless, a key benefit for the community is that the company has undertaken a few corporate responsibility projects: cooperating with village authorities, they sponsored different village projects such as installation of a village water system and education necessities (desks and chairs). The company also makes its dispensary available to workers

and local communities surrounding the farm, thus boosting available medical services in the area.

While smallholder farmers, food vendors and traders in other parts of Tanzania were hard hit by the COVID-19 pandemic despite minimal lockdown provisions [19,107], our case study area did not experience as many difficulties. In the study area, many small-holders said they did not feel they had experienced any impacts, because they carried on working for the company throughout the pandemic. Those who did experience difficulties could not easily access farming equipment and agricultural inputs, with dwindling supplies and new supplies having delayed delivery.

## 5. Discussion

The case studies share some similarities with large-scale agricultural investments across the continent, since investors have established large estates and employed locals in their businesses. However, the case studies differ markedly in terms of how the land investment deals were negotiated. In the case of Tanzania, locals did not know how the company had acquired land without consultation, revealing that the VGGT clauses about consultation were ignored in this case. By contrast, in Mozambique, CAD negotiated with a local farmers' association, with the support of the local government. Table 1 below highlights some similarities and differences between the case studies.

**Table 1.** Differences in the structure and benefits in the case studies.

| Criteria | China Africa Development Company (Mozambique) | China State Farm Agribusiness (Tanzania) |
|---|---|---|
| Large estate | Yes | Yes |
| Negotiation with communities to acquire land | Some | No |
| Environmental and social impact assessment | No | No |
| Legally binding agreement | Agreement not legally binding | No |
| Created local employment | Yes, but only low-paid, poorly educated works | Yes |
| Corporate social responsibility | No | Yes |
| Impact on food security | Some benefitted (cheaper food), some were negatively impacted (unable to sell at a good price at the market) | Unclear |
| Conflict | (a) Yes, Association members did not share viewpoints and some left Association (b) Some community members feared CAD might acquire more land | Land shortage in the area, compounded by the investment, leads to conflict |

In the case of China State Farm Agribusiness (Tanzania), it was clear that the company had not applied VGGT principles when pursuing its investment, in that the company had not negotiated with the people occupying the land before proceeding with the project. Further, despite calls from the community to negotiate to use the land that the company was not using, the company had not kept communication open or revisited the possibility for re-negotiation. CAD in Mozambique had also not fully fulfilled its responsibilities in terms of the VGGTs, because even though some negotiation had taken place and some agreements had been made, none of this was binding. Therefore, many community members felt they had not been consulted, since no participatory process had taken place. In the view of several Association members, the company was not fulfilling the terms of the agreement that had been negotiated, confirming previous research showing that the way

Mozambican smallholder associations are a success in negotiating benefits in large-scale agricultural investments, these claims are overstated [108]. Both these examples highlight how important it is for investors to ensure that negotiations are fully participatory and that binding agreements are made.

Further, a once-off negotiation is insufficient for ensuring that communities accrue sufficient benefits for the loss of their land; investors need to remain open to revisiting the conditions of the agreement if many community members are experiencing negative effects. Without the possibility for further communication, conflicts can arise among community members due to variations in who wins and who loses. For example, in the Tanzanian case, land shortages—not solely, but partly, resulting from the investment—were creating sometimes violent conflicts in the area. In Mozambique, conflict over the CAD investment had led to most Association members leaving the Association, as they were unhappy with the lack of benefits being provided, which they regarded as the company reneging on the agreement.

Although many negatives were experienced in both communities, some benefits did accrue for some community members. For example, some of the infrastructural work undertaken by CAD in Mozambique led to some community members being able to access more land. As we could not visit the site, we could not establish how many community members benefitted from this and what the demographics were for those who benefitted. However, previous research on large-scale land-based investments has shown that they result in social differentiation in terms of who wins and who loses out [23,86–92]. The interviews with Association members in this community are indicative of such social differentiation occurring within the CAD area, since some members viewed the investment as positive, while others left the Association because they did not perceive sufficient benefits from the project. This is especially the case in respect of the lack of CSR projects to counter-balance the loss of land.

In contrast to CAD, some interviewees in the Tanzanian case perceived CSR as a benefit, albeit they did not elaborate on, nor could we establish the extent of, these projects. For example, it was unclear how many desks and chairs had been provided to the school, which community members were able to access the water system infrastructure provided, and whether all/most or only a few were benefitting from the company dispensary. This warrants further investigation.

With respect to another positive spin-off identified by community members in both cases, employment was perceived as a win. However, in the case of CAD in Mozambique, the fact that company policy only supported the employment of less-educated community members in order to keep wages down reveals how class comes into play in terms of who wins and who loses in large-scale land-based investments. While it could be argued that CAD was preferentially targeting the most vulnerable community members to offer employment, the low wages on offer counter-balance the possibility of the most vulnerable community members moving up the social ladder. Therefore, even though CAD had an opportunity to develop more equitable social relations in the community, it did not achieve this.

By contrast, the China State Farm Agricultural Company in Tanzania provided job opportunities with more benefits, including on-site accommodation for many workers. However, without being able to access more information about the employment conditions, it is difficult to say who benefitted and who lost out. For example, if those employed by the company included managers from the local area, they would likely have benefitted more from the investment and perhaps had better working conditions. However, some of the workers complained about the poor working conditions and the threat of losing their position. To confirm the levels of social differentiation in working conditions and incomes, we would have needed to visit the site and also talk to locals in management positions at the company. This is an important next step in this research.

The negative impacts of COVID-19 on agriculture in the case study areas were similar to the findings in other studies, discussed in the literature review above. From the literature,

we identified common features of the impact of COVID-19 on agriculture, and these are shown as variables in Table 2. Table 2 also shows the extent to which these variables came into play in the two case study areas.

**Table 2.** Different impacts of COVID-19 in case studies.

| Issue | China Africa Development Company (Mozambique) | China State Farm Agri-business (Tanzania) |
|---|---|---|
| Disruption in access to inputs | Yes | Yes |
| Disruption in access to markets | Yes | No |
| Difficulty accessing food | Yes | No |
| Support from investors during the crisis | No | No |
| Protective measures taken by government/development partners | Yes | No |
| Loss of land tenure | No | No |

Although the research from elsewhere in Tanzania showed that some areas experienced many difficulties associated with COVID-19 [57,61,93,94], in our case study, most interviewees did not experience any major disruptions to their lives. This can be attributed to the minimal measures imposed by the Tanzanian government to slow the spread of the disease [109], such that in this community, people still had freedom of movement, they were still employed, and they still had access to markets. However, some smallholders experienced difficulty accessing farm inputs, with the knock-on effect that they could not plant in time, which could have negatively impacted medium-term production as they would not be able to produce enough for subsistence or to sell. However, it would be necessary to follow up further to assess the extent to which this impacted them, and the extent to which the impacts were based on social differentiation, which other studies have found to be a feature of COVID-19 impacts on agriculture [110].

The main impacts in Mozambique also related to input supplies; however, due to interventions by government and development agencies, which targeted the most vulnerable—including women—this area did not experience severe consequences. Further, the provision of irrigation to some smallholders, mainly women, likely helped boost production. Nevertheless, since not all smallholders received the inputs, differential impacts were likely felt across the community, which requires further investigation. Because of the supply of some inputs, the key problem incurred by many of the farmers was that they could not access markets to sell their produce. CAD could have stepped in to help with this issue when transporting its own produce to market; however, the community did not experience any support from the company. The overall lack of CSR support both prior to and during the pandemic points to a problem with how investment was undertaken in this case, as CAD did not seem to recognise any responsibility toward supporting the community from which it had acquired land.

## 6. Conclusions

The case studies show that, firstly, even when Chinese investors undertook negotiations with local communities, it was not sufficiently broad-based to ensure that communities in Mozambique and Tanzania felt they had been consulted. Secondly, in both case studies, community members were frustrated about the ongoing lack of communication between themselves and the company. Despite a few positive spin-offs, both cases reveal that more work is needed to resolve existing conflicts due to investors' ongoing poor engagement with communities in the surrounding areas: in the case of Tanzania, this revolves around the community's need to access land that the company is not using, and in the Mozambican case, it revolves around the need for more CSR projects. This highlights how, even if VGGTs are used to guide investments, close scrutiny must be given not only to land tenure issues

during the initial negotiation, but also to ensure that ongoing engagements include CSR responsibilities and labour rights, and these are entrenched in agreements. Indeed, as has been shown in previous studies [94,95], these case studies point towards local communities' contestations of land-based investments when they are not directly empowered to participate and instead their decision-making capacities are limited. The cases also manifest documented complexities and impacts of land-based investments, which create groups of winners, who have education, larger plots, higher incomes and secure high-level jobs, and losers, who have marginal or small plots of land and are employed in casual or temporary jobs, as documented in other studies [111–114].

Similarly, the COVID-19 pandemic raised issues about CSR in times of crisis. In Mozambique, CAD did not provide any support to the surrounding community, further upsetting community members already frustrated with the lack of CSR before the crisis. While the Tanzanian community did not experience serious fallout from the pandemic, questions need to be asked about the safety measures taken by the company, since workers worked through the pandemic. Overall, this global crisis points to the need for VGGTs to include provisions on companies' direct responsibilities in such situations, since the pandemic has highlighted how locals and investors could better collaborate to achieve inclusive and sustainable economic recovery, especially with regard to equitable access to resources for men and women, including agriculture, land and natural resources.

Finally, the findings suggest that both the Tanzanian and Mozambican governments need to do more to integrate VGGTs in their policies and laws. While Tanzania has, since November 2017, established the Technical Cooperation Programme (TCP) to further support VGGT implementation in the country, there is no or little information about what activities are in place or carried out in Mozambique to raise awareness and also implement VGGTs in the country.

*Limitations of Study*

A key limitation of the study is that COVID-19 meant we could not make site visits to confirm some information highlighted by the interviewees. For example, in the case of Mozambique, we could not examine the area in which interviewees said that the Chinese company had opened the valleys, so we could not establish how many smallholders benefitted from the land that had been opened. We also could not verify the prices CAD and smallholders received across the value chain and at different markets, nor could we assess whether the quality of produce from smallholders and CAD was variable. In Tanzania, we could not verify the number of hours that employees were working at the China State Farm Agribusiness. We also could not confirm the quality and quantity of corporate social responsibility (CSR) projects that had taken place and the extent to which they were functioning. For example, we did not have an opportunity to visit the company dispensary to investigate the level at which the community was using the facility. All these factors meant that we had to heavily rely on interviews, even though more verification would have strengthened the study.

**Author Contributions:** Conceptualization, E.S. and R.P.; data curation, E.S., R.P. and C.N.; funding acquisition, E.S.; methodology, E.S.; writing—original draft, E.S., R.P. and C.N.; writing—review & editing, R.P. and E.S. All authors have read and agreed to the published version of the manuscript.

**Funding:** The work was executed under Letter of Agreement between FAO and the Tanzania Land Alliance (TALA) under project GCP/INT/328/UK with support from the Government of Great Britain and Northern Ireland (Department for International Development (DfID)).

**Institutional Review Board Statement:** Not applicable.

**Informed Consent Statement:** All our interviewees provided verbal consent before interviews.

**Data Availability Statement:** Not applicable.

**Acknowledgments:** We thank FAO and Bernard of TALA for their support during this study and we are grateful to all the people we interviewed during this study.

**Conflicts of Interest:** The authors declare no conflict of interest.

## Abbreviations

| | |
|---|---|
| APEIX | *Agencia de Promoção de Exportações e Importações* |
| BRELA | Business Registration and Licensing Authority |
| CAD | China Africa Development Company (Mozambique) |
| CFS-RAI | Committee on World Food Security Principles for Responsible Investment in Agriculture and Food Systems Responsible Investment in Agriculture and Food Systems |
| DNT | National Directorate of Land |
| DUAT | *Direito de Uso e Aproveitamento da Terra* |
| ESIA | Environmental and Social Impact Assessment |
| GRO | Granted Right of Occupancy |
| MOU | Memorandum of Understanding |
| NEMC | National Environment Management Council |
| OSHA | Occupational, Safety and Health Authority |
| PEDSA | Strategic Plan of Development of Agriculture Sector |
| PNISA | National Agriculture Investment Plan |
| PROAGRI | National Development Programme |
| SPGC | Provincial Service of Geography and Cadastre |
| TANESCO | Tanzania Electric Supply Company Limited |
| TBS | Tanzania Bureau of Standards |
| TCP | Technical Cooperation Programme |
| TFDA | Tanzania Food and Drugs Authority |
| TIC | Tanzanian Investment Centre |
| TRA | Tanzania Revenue Authority |
| VGGTs | Voluntary Guidelines for the Responsible Governance of Tenure of Land, Forests and Fisheries in the Context of National Food Security |

## Appendix A

Open-ended questions asked of farmers' association members, farmers that did not belong to farmers' associations, other community members, and where applicable, company workers.

1. What do you know about how the land investment came about?
2. Who participated in negotiations for the investor to acquire land?
3. Was the negotiation process satisfactory?
4. How did the investment affect your land tenure?
5. Now that the investor is working in the area, what are the key benefits you can identify? How has the company been supportive?
6. Now that the investor is working in the area, what are the key problems you can identify? How could the investor be more supportive?
7. Did the investor meet your expectations and/or the terms of the agreement?
8. How did COVID-19 impact your farming activities?
9. How did COVID-19 impact your access to food?
10. Did they give additional support during COVID-19?

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
