# Peer review of "Smallholder Views on Chinese Agricultural Investments in Mozambique and Tanzania in the Context of VGGTs"

_sustainability, doi:10.3390/su15021220_

Round 1

Reviewer 1 Report

Dear authors, please find my comments below: I hope this will assist in improving the quality of the study.

1.             The abstract needs to be revised in light of the comments below and should contain all parts (background, methods, results, and conclusion) in an unstructured way.

2.             Authors should discuss the implications of COVID-19 on small farmers’ introduction because it is also one of their key study objectives.

3.             COVID-19's impact on agriculture should also be part of the literature review section.

4.             Lines 77–81 should be part of "Materials and methods" under the subheading "Data collection."

5.             How did you select the sample sizes in both countries? Please briefly explain under the subheading "sampling methods."

6.             From where did you get lists and phone numbers of farmers in both countries?

7.             Did you use any data collection instruments? If yes, please explain what type of questionnaire it was. It all should be part of your "materials and methods" section.

8.             For the easy understanding of readers and to make comparisons between two countries easier, authors need to present some of the results in tabular or graphical form for both countries.

9.             It looks like there is no difference between the discussion parts and the results. Authors need only present results in the "Result" section. 

10.         Moreover, discussion and conclusion should be separated.

11.         Please mention the limitations of the study in the last paragraph of the "discussion or conclusion" section.

12.         References in the text should be changed according to journal style.

Wishing you good luck

Author Response

Thank you to the reviewer for your insightful comments. They have certainly helped us improve the paper and correct some oversights in our approach. Accordingly, we have provided additional information and restructured the paper.

Changes are extensive, as can be seen in the redlined copy of the paper: we hope this is satisfactory. With regard to your specific points, please see our replies below:

  1. Added: sentence highlighting our investigation of Covid-19, sentence highlighting the purpose of the VGGTs, and the conclusions, including conclusions about Covid-19
  2. Included one sentence on Covid-19 in the introduction
  3. Added a paragraph on Covid-19 in the literature review
  4. Moved the sentence and created subheadings in the methodology section
  5. Added - snowball
  6. Through literature review and snowball
  7. Open-ended questionnaires - added mention to methodology and added questions in the Appendix)
  8. Two tables inserted: one comparing issues in the cases and the second on Covid-19 in each community
  9. Separation made
  10. Separation made
  11. Including limitations in conclusion
  12. Changed references with square brackets and not superscript.

All changes have now been highlighted in yellow.

Reviewer 2 Report

The general opinion about the article: Smallholder Views on Chinese Agricultural Investments in Mozambique and Tanzania

This paper might be a review paper, not an article because it lacks basic scientific methodology, data processing, and sound scientific results. 

It has no proper sample, and no appropriate methodology.  

In the abstract the authors mentioned: Telephonic interviews with 20 smallholders in Mozambique and 35 in Tanzania…

The sample size and method of data collection are far too small to create a scientific article, therefore it may be considered as a review because it is written in a very descriptive way.

If the authors declared in line 112: More than 2.5 million hectares of general land have been identified as suitable for investors… then the sample is statistically not viable nor are findings significant.

The descriptive sense of writing is shown e.i. in lines 492-494„Farmers’ approaches directly to the company have not led to the expected level of support; instead, the district government in Matutuine District has intervened several times and persuaded CAD to apply corporate social responsibility (CSR).

A further conflict is because CAD sell some of their produce in the local market at 495 cheaper prices, so some local vendors are losing customers

There are no statistical or research results provided to support these hypotheses

And onward lines 505-510: Although many criticisms have been leveled at the company, many community members are still pro-investment, and the local government and some private investors  consider this as a good example of land investment in the country. Some farmers said they  had benefitted from investment in the area. For example, because the company uses a lot  of water, the valleys were opened and communities can now.

There are no statistical nor research results provided to support these hypotheses, no tables, no correlations, no Anova or other analysis

And further in lines 574-576 no evidence no statistical nor research results were provided to support these hypotheses, no tables, no correlations, no Anova or other analysis

However, employees also complain of tough rules, with long working days and their productivity being closely monitored; if Chinese staff say that they are not meeting their quotas, they risk having their salary reduced.

The authors should follow the technical guidelines about the references; line them all in brackets [1], not as set in text with exponent numbers1,2

Author Response

Thank you to the reviewer for your insightful comments. They have certainly helped us improve the paper and correct some oversights in our approach. Accordingly, we have provided additional information and restructured the paper.

Changes are extensive, as can be seen in the redlined copy of the paper: we hope this is satisfactory. With regard to your specific points, please see our replies below:

  1. We have updated the methodology and presented the key results in tables
  2. Sampling process added - snowballing
  3. The sample size is not based on the area of the country covered by investments; the sample is based on the number of Chinese investments in each country - which are few. Based on these few investments in each country, we selected the case studies areas.
  4. The sample size is small due to the constraints of accessing telephone numbers and undertaking telephonic interviews. We added this info to the limitations of our study in the conclusion
  5. We have undertaken more analysis - not just description, and this is included under the heading "Discussion" as well as in the conclusion
  6. Due to the limitations of moving around during Covid-19 we could not go to relevant markets to determine the market price. Further, the research is based on the views of the community members - and their view is that prices and quality differ.
  7. We could not visit the area to gather photographic evidence of the area of land opened or to establish the exact size and establish how many farmers benefitted from the land being opened - unfortunately, Covid-19 really restricted the information we could access
  8. Reference style changed.

Round 2

Reviewer 1 Report

Thank you for addressing my comments.

Author Response

Dear reviewer

Please find attached the file with yellow highlights showing the changes we made.

Thanks again for your incisive comments.

Reviewer 2 Report

major revison equired

Author Response

Dear Reviewer

Thank you very much for your insightful comments which we have endeavoured to answer, where possible. Please note that on the file, we have highlighted changes in yellow so you can easily scrutinise the changes. Below find our responses:

  1. We have revised our report to include a more detailed methodology. In terms of the results, we have elaborated the significance in the section called Discussion
  2. Sample method: now included
  3. Sample size: Given the constraints of not being able to visit the case study area, our sample is on the small side as we could not reach more people telephonically, however, we spoke to many key community members, and feel that the sample is adequate under the circumstances.
  4. While Mozambique has set aside more than 2.5 million hectares of general land for investors (which could be investment in many things besides agriculture) our study ONLY focuses on Chinese agricultural investors. With respect to Chinese agricultural investments, we only found five other investments in Mozambique and six other investments in Tanzania; therefore, we regard one case study from each country a significant sample.
  5. The results are presented in a descriptive way because this is a qualitative study - not a quantitative study. In a qualitative study, it would be unusual to have statistics
  6. However, we have created tables  and discussed the significance of the two cases, comparing and contrasting them in the (added) Discussion section, as well as in the Conclusion
  7. We could not gather statistical information on prices on offer at the market, as due to Covid-19 we could not visit the market. Therefore, we have to rely an the qualitative data - and after all, this is a qualitative study
  8. We removed the reference to the views of local government and investors as this was part of a much larger study
  9. We could not visit the area, so we do not know amount of land opened by the company, and we also do not know who benefitted from this, how many people benefitted, and what size of land they were each able to access. However, we believe those who benefitted would see it as an advantage, so this meets the merits of qualitative data.
  10. There are no statistics because this is not a statistical study: it is a qualitative study.

I hope these answer suffice.

Kind regards

Round 3

Reviewer 2 Report

Dear authors,

some of the comments made were followed, some not

in the methodology the questionnaire was mentioned,

but you do not explain which questions you have made, from where they were used from, you are sure not the first in history to use a questionnaire and questions

no reasons were given for line 399 :Based on an extensive literature review, we identified two case studies for in-depth investigation the China Africa Development Company (CAD) ..." no literature was given

state some references

best regards

the reviewer

Author Response

Thank you again for the helpful comments. Please see replies below:

  1. I am sorry, I did provide a reference to the questionnaire in text saying:  

    "shown in the Appendix to this article". I have now highlighted this text in the attached in blue so that it's clearer and I also highlighted the Appendix at the end of the article in blue.

    .
  2. Added citations and references: highlighted in blue.

Kind regards
